# Small Extracellular Vesicles and Oxidative Pathophysiological Mechanisms in Retinal Degenerative Diseases

**DOI:** 10.3390/ijms25031618

**Published:** 2024-01-28

**Authors:** Francisco J. Romero, Manuel Diaz-Llopis, M. Inmaculada Romero-Gomez, Maria Miranda, Rebeca Romero-Wenz, Javier Sancho-Pelluz, Belén Romero, Maria Muriach, Jorge M. Barcia

**Affiliations:** 1Hospital General de Requena, Conselleria de Sanitat, Generalitat Valenciana, 46340 Requena, Spain; romero_rebwen@gva.es; 2Facultad de Medicina y Odontología, Universitat de València, 46010 Valencia, Spain; manuel.diaz@uv.es; 3Facultad de Ciencias de la Salud, Universidad Europea de Valencia, 46010 Valencia, Spain; mariainmaculada.romero@universidadeuropea.es; 4Facultad de Ciencias de la Salud, Universidad CEU-Cardenal Herrera, 46115 Alfara del Patriarca, Spain; mmiranda@uch.ceu.es; 5Facultad de Medicina y Ciencias de la Salud, Universidad Católica de Valencia, 46001 Valencia, Spain; fj.sancho@ucv.es (J.S.-P.); belen.romero@ucv.es (B.R.); jm.barcia@ucv.es (J.M.B.); 6Unidad de Cuidados intensivos, Hospital de Manises, 46940 Manises, Spain; 7Facultad de Ciencias de la Salud, Universitat Jaume I, 12006 Castelló de la Plana, Spain; muriach@uji.es

**Keywords:** exosomes, retinal diseases, oxidative stress, autophagy, microRNAs

## Abstract

This review focuses on the role of small extracellular vesicles in the pathophysiological mechanisms of retinal degenerative diseases. Many of these mechanisms are related to or modulated by the oxidative burden of retinal cells. It has been recently demonstrated that cellular communication in the retina involves extracellular vesicles and that their rate of release and cargo features might be affected by the cellular environment, and in some instances, they might also be mediated by autophagy. The fate of these vesicles is diverse: they could end up in circulation being used as markers, or target neighbor cells modulating gene and protein expression, or eventually, in angiogenesis. Neovascularization in the retina promotes vision loss in diseases such as diabetic retinopathy and age-related macular degeneration. The importance of micro RNAs, either as small extracellular vesicles’ cargo or free circulating, in the regulation of retinal angiogenesis is also discussed.

## 1. Introduction

‘The retina is the brain’s window on the world’, as it appears in the magnificent neural sciences compendium of Kandel and others [1], and as such, it is regarded as part of the central nervous system. The retina has several cell types, among which are two types of photoreceptors, cones and rods, distributed asymmetrically: cones mostly in the center (*macula lutea*) and rods mostly in the periphery. Neurons that form the optic nerve, retinal ganglion cells, are also in the retina together with other nerve cell types: bipolar cells, horizontal cells, amacrine cells and Müller cells [2]. Since half of the cerebral cortex is used to process the signals arriving from approximately one million fibers of the optic nerve, the visual information coming from the retina is highly valued; thus, visual impairment and blindness are among the most feared disabilities of mankind. The retinal pigment epithelium (RPE) plays an important role in maintaining the function and the integrity of the retina and choroid [3]; it absorbs light and protects against photooxidation [4]. RPE is part of the blood–retinal barrier, and it regulates the entrance of nutrients into the retina. In fact, RPE transports nutrients, ions, water, and metabolic end products from one side of the retina to the other [5]. RPE is critical for phototransduction, is also responsible for phagocyting the tips of photoreceptors’ outer segments, and secretes a number of growth factors essential for the structural integrity of the retina and the choroid. Thus, it has been repeatedly involved in the pathophysiology of major retinal diseases such as age-related macular degeneration (AMD) [6] and diabetic retinopathy (DR) [7].

Photoreceptors and retinal ganglion cells face heightened susceptibility to oxidative stress, attributed to elevated oxygen levels, glucose oxidation, and the abundance of polyunsaturated fats in lipids. This vulnerability is exacerbated by the phototransduction process, resulting in an augmented production of reactive oxygen species (ROS). Numerous retinal diseases have been linked to an imbalance in ROS, as evidenced in various studies [8,9,10,11]. Primary reactive species in this context encompass the superoxide anion, singlet oxygen, hydroxyl radical, hydrogen peroxide, peroxynitrite, and nitric oxide. Moreover, the oxidation of polyunsaturated fatty acids can give rise to aldehydes like malondialdehyde or 4-hydroxy-alkenals, recognized as distinctive markers of oxidative stress, with the latter also possessing signaling properties [12].

Our interest in retinal oxidative pathophysiology started in the early 90s, when two of us (M.D.-L. and F.J.R.) joined efforts to describe a very relevant part of the antioxidant system of the retina in human donors: the glutathione system [13]. Although diseases affecting the anterior segment of the eye are the primary causes of vision impairment and blindness globally (cataract, refractive error, and glaucoma), DR and AMD are also among those, especially in the elderly population and developed countries (https://www.who.int/news-room/fact-sheets/detail/blindness-and-visual-impairment, accessed on 2 January 2024). In these nations, retinitis pigmentosa (RP), a group of inherited forms of retinal degeneration, is a prevalent cause of blindness in the working-age population [14]. Oxidative stress has been implicated in the pathogenesis of RP and antioxidants clearly delay RP progression, at least in experimental models [9].

The need to maintain retinal homeostasis for adequate vision and the indisputable role that redox equilibrium plays in the pathophysiological mechanisms in so many retinal diseases have received constantly growing interest in the last decades. Our aim in this manuscript is to review the pathophysiological mechanisms related to or regulated by oxidative burden in retinal diseases, with specific emphasis on the role of angiogenesis and extracellular vesicles and their cargo.

## 2. Oxidative Pathophysiological Mechanisms

As mentioned above, this review focuses on pathophysiological mechanisms that have been clearly related to or are regulated or affected by oxidation, where antioxidants, by different mechanisms, may exert beneficial effects. The angiogenesis mechanism as part of the pathophysiology of vision loss in retinal diseases has been already introduced above. The need for cellular communication in this process, and certainly others, led us to study in detail the phenomena involved in it. Within multicellular organisms, cellular communication relies on extracellular signaling molecules, including nucleotides, lipids, and proteins. Following their release into extracellular medium, these molecules interact with receptors on neighboring cells, initiating intracellular signaling pathways and influencing the physiological state of the recipient cell. Beyond the conventional release of signaling molecules, eukaryotic cells possess the ability to release vesicles into the extracellular medium. These vesicles encapsulate various types of molecules, including microRNA (miRNA) sequences. Remarkably, miRNAs within these vesicles exhibit regulatory actions in diverse retinal cells.

### 2.1. Extracellular Vesicles

Extracellular vesicles (EVs) encompass a diverse array of proteins, lipids, and nucleic acids that hold the potential to influence the fate of target cells [15]. These vesicles play a pivotal role in cell communication, facilitating the transfer of information, including proteins, mRNA, miRNAs, and even DNA fragments, among cells [16]. Notably, EVs contribute to essential processes such as immune suppression of tumor metastasis, angiogenesis, and autophagy [17,18,19,20,21]. The release of EVs is influenced by the cellular environment [22,23,24,25]. These vesicles can be categorized based on size as either sub-micro sized, referred to as microparticles, or nanometric sized, previously termed exosomes and currently recognized as small extracellular vesicles (sEVs) [26,27,28]. Typically, sEVs carry a cargo comprising various molecules, including proteins, lipids, and genetic material such as miRNA [29,30]. The discernible correlation between the composition of sEVs (proteins, lipids, mRNAs, and miRNAs) and the cell type, as well as the physiological state of the cell, has prompted the proposal of sEVs as potential biomarkers for diagnosing various human diseases [31,32], as well as those of the visual system [33], where their role in normal and diseased eyes has been reviewed [34].

It is generally accepted that sEV production is stimulated by oxidative stress; Malik and collaborators showed that 2 h ischemia followed by either 1 h reperfusion or ethanol was able to stimulate sEV production in cardiac myocytes [35]. They showed that ethanol-induced sEVs appeared to have an increased protein content compared to hypoxia- and reoxygenation-induced sEVs, based on similar acetylcholine-esterase activity, which reflects the amount of membrane present. Furthermore, after two hours of ethanol exposure, a dramatic increase in ROS in cardiac myocytes was also observed. Interestingly, after treatment with antioxidants to reduce ROS production, decreased sEV formation was obtained. A similar approach with retinal pigment epithelial cells also confirmed that under oxidative conditions, the number of sEVs produced by these cells was increased and their cargo also varied [22,36]. A very innovative approach using human primary RPE cultures from human induced pluripotent stem cells (hiPSC)-derived retinal organoids has recently described that sEVs released from these cells under oxidative burden contain drusen-associated proteins [37]; this represents a new insight into the pathophysiology of AMD that certainly confirms a crucial role for sEVs in this process, but also for drusen-associated proteins, since it has been brilliantly challenged that drusen do not directly produce visual deterioration [38].

Other authors have used rotenone to challenge ARPE-19 cells and compared the effects of the sEVs obtained from these cells with the effects of those obtained from control cells. sEVs released by rotenone-stimulated ARPE-19 cells have demonstrated the ability to induce cell apoptosis, oxidative injury, and inflammation in control ARPE-19 cells. Moreover, these same sEVs, secreted under oxidative stress (OS) conditions, exhibit the capacity to impair retinal functions in rats while upregulating the expression of Apaf1. The elevated expression of Apaf1 in sEVs produced under OS conditions leads to the inhibition of cell proliferation, an increase in cell apoptosis, and the initiation of an inflammatory response in ARPE-19 cells, operating through a caspase-9 apoptotic pathway [39]. In a mouse model of retinal degeneration induced by prior intraperitoneal injection of MNU (Methyl-Nitroso-Urea), subretinal injection of sEVs derived from ARPE-19 cells was found to alleviate visual function impairments and structural damages in the retina of MNU-treated mice. Furthermore, RPE-derived sEVs demonstrated positive effects on the electrical response of inner retinal circuits, suppressed the expression levels of inflammatory factors, and mitigated oxidative damage in MNU-treated mice [40]. Co-cultures of ARPE-19 (of human origin) and primary porcine RPE cells in a transwell model, stimulated with hydrogen peroxide, produced sEVs that contained HDAC6, which is known to reduce tight junction stability, action that was dependent on sEV uptake; this led these authors to propose that sEVs are able to communicate stress messages to healthy RPE cells, eventually contributing to RPE dysfunction [41].

Kannan and collaborators have recently reviewed the role of alpha crystallins in the RPE and their implications for the pathogenesis and treatment of AMD [42]. This research group provided evidence that under severe oxidative conditions, RPE released alpha crystallin via the exosomal pathway and that it accumulated in the basolateral side of RPE, once the barrier function has broken down, suggesting sEVs are a source of alpha crystallin for the drusen [43]. α-Crystallins, prominent members of the small heat shock protein family, extend beyond their well-recognized chaperone effect, showcasing a diverse range of properties. These include anti-inflammatory, antifibrillar, antiapoptotic, protection against ER stress and autophagy, and modulatory effects on angiogenesis [44]. It is noteworthy that studies investigating the antiapoptotic function and associated signaling mechanisms of α-crystallins often utilize oxidative stimuli as a model. Notably, RPE cells overexpressing either αA or αB crystallin exhibit increased cellular glutathione (GSH) content. Additionally, researchers have demonstrated that a selective increase in the mitochondrial GSH compartment in oxidatively stressed RPE cells overexpressing αA and αB provides cellular protection, aligning with the proposed role of the mitochondrial GSH pool in various cell types [45]. This perspective gains significance in light of the suggestion that mitochondrial DNA damage plays a role in the pathophysiology of AMD [46]. Interestingly, these sEVs have also been associated with other eye diseases such as glaucoma. Recent research has elegantly confirmed that enriched sEVs, obtained under hypoxic conditions from human amniotic membrane stem cells and containing a plethora of functional molecules, exhibit therapeutic potential in glaucoma, by shielding retinal cells against oxidative and hypoxic injuries in vitro and by promoting the recovery of intraocular pressure (IOP) and mitigating retinal degeneration in vivo [47].

Using an experimental approach with primary polarized RPE cultures under subtoxic oxidative stress conditions, changes in sEV proteins content involved in epithelial barrier integrity have been reported, which include basal-side specific desmosome and hemidesmosome shedding via sEVs [48], which would explain the RPE-impaired outer blood–retinal barrier function that occurs in AMD. Cellular oxidative damage of the RPE cells induced by NaIO3 could be prevented by the administration of sEVs obtained from mesenchymal stem cells [49], that decreased the levels of reactive oxygen species, and upregulated SOD activity by a mechanism that has been ascribed to the up regulation of the expression of the Nrf2 pathway, since Nrf2 inhibitors block this antioxidant effect [49]. Moreover, recent findings indicate that under oxidative conditions, RPE cells upregulate the expression of CYLD-AS1, a long, noncoding mRNA with the ability to modulate the expression of Nrf2 (associated with oxidative stress) and members of the NF-κB pathway, which play a role in inflammation. Notably, the regulation of these two signaling pathways by CYLD-AS1 involves interaction with miR-134-5p. Interestingly, sEVs released by RPE cells with CYLD-AS1 knockdown exhibited a reduced proinflammatory effect compared to those secreted by control cells [50]. Based on these observations, the authors suggest that targeting CYLD-AS1 could be a potential therapeutic approach for treating AMD.

It has been reported that the complement regulatory protein (CD59) was increased in regions of the RPE of early AMD patients, but decreased in advanced forms of AMD [51]. Dr. Handa’s group showed that this protein was released within sEVs to the subretinal space [52]. The capability of RPE-released sEVs to induce angiogenesis in an experimental model has also been demonstrated, proposing for the first time the hypothesis of sEVs as relevant players in retinal neovascularization [36].

### 2.2. Angiogenesis

Angiogenesis, the process of forming new blood vessels from existing vasculature, is responsive to various stimuli. Two distinct types of angiogenesis exist: physiological and pathological. Physiological angiogenesis occurs during embryonic vasculature growth and persists in early postnatal development, being crucial for supplying oxygen and essential nutrients to growing organs. In contrast, pathological neovascularization is characterized by uncontrolled and disorderly vasculature growth, manifesting when there is an imbalance between angiogenic stimulators and inhibitors. The initial characterization of this process was made by Judah Folkman in 1971 [53]. Angiogenesis involves two primary cell types: pericytes and endothelial cells. Pericytes contribute to vascular stability and enhance endothelial cell barrier function through direct contact and paracrine regulation, as comprehensively reviewed elsewhere [54]. Pericytes, activated by the hypoxia-inducible factor pathway under hypoxic conditions, can release sEVs that modulate endothelial cell migration, sprouting, and angiogenesis in models such as wound healing and spinal cord explant cultures [55]. Consequently, pericytes, along with sEVs from neurons, glia, and endothelial cells, in addition to circulation, play crucial roles in regulating endothelial cell integrity and intercellular crosstalk within the neurovascular unit under both physiological and pathological conditions [56,57,58].

Abnormal angiogenesis allows the extravasation of plasma proteins, establishing a provisional matrix, necessary for the migration of activated endothelial cells. Basal membranes and the extracellular matrix are degraded locally by metalloproteases, allowing the underlying endothelial cells to migrate following the trail of the angiogenic stimulus into the perivascular space [59]. Another study [60], not originally designed to look for retinal pathophysiological mechanisms but for the microvascular damage in diabetes, obtained results that fit again in demonstrating that sEVs from adipocytes obtained under high glucose conditions, could adversely affect the environment of microvascular endothelial cells through a LINC00968/miR-361-5p/TRAF3 signaling pathway [60].

Hajrasouliha’s research group has explored the hypothesis that cells within the neurosensory retina release sEVs capable of regulating angiogenesis. Their observations revealed that sEVs derived from retinal astrocytes in normal mice contain multiple antiangiogenic factors, effectively impeding the development of choroidal neovascularization in a laser-induced model. In contrast, sEVs derived from RPE did not exhibit the same antiangiogenic properties. These findings suggest that in the retina, sEVs from different cell types may play a role in balancing anti- and proangiogenic signals. The mechanisms through which sEVs from retinal astrocytes exert their effects involve the suppression of retinal vascular leakage and the inhibition of choroidal neovascularization in this laser-induced mouse model. These effects may be attributed to the ability of retinal astrocyte-derived sEVs to target both macrophages and vascular endothelial cells. Infiltrating macrophages are significant contributors to inflammatory cytokines, complement, and VEGF, all of which are crucial in inflammation and new vessel formation. Retinal astrocyte-derived sEVs were found to suppress the infiltration and migration of macrophages in a chemotactic chamber. Furthermore, these sEVs inhibited the migration of endothelial cells and the formation of vascular tubules. The antiangiogenic impact of retinal astrocyte-derived sEVs on macrophages and endothelial cells is likely facilitated by their cargo, which includes proteins, lipids, mRNA, and miRNAs. The analysis with an antibody array of retinal astrocytes-derived sEVs revealed the presence of molecules with anti-angiogenic properties such as endostatin, PEDF, and TIMP-1. However, several chemokines and metalloproteases were detected in these sEVs that can either promote or inhibit angiogenesis, depending on their environmental context [61].

The role of RPE cell-derived sEVs in the pathogenesis of dry (especially after the findings by Flores-Bellver and coworkers [37] mentioned above), and wet AMD cannot be ignored. RPE cell-derived sEVs may directly promote the occurrence and development of choroidal neovascularization in wet AMD. It has been demonstrated that oxidative burden induces RPE cells to secrete more sEVs, and that these sEVs effectively transmit signals that are able to induce tube formation in endothelial cells [22]. Moreover, it has been demonstrated that stressed RPE cells release a higher quantity of sEVs compared to control cells. These stressed RPE-derived sEVs exhibit elevated expression of VEGF receptors (VEGFRs) on their membranes and contain an additional cargo of VEGFR mRNA. Angiogenesis assays conducted in this context confirmed that endothelial cells display an increased capacity for tube formation when exposed to stressed RPE-derived sEVs [22]. Subsequent experiments established a connection between two phenomena: autophagy and sEVs release in RPE cells under oxidative conditions. These two cellular mechanisms were found to contribute to the regulation of angiogenesis [36]. In a normal physiological environment, there exists a delicate balance between angiogenic activators and inhibitors, leading to limited new blood vessel formation. However, various visual conditions, such as age-related macular degeneration (AMD) and diabetic retinopathy (DR), can disrupt this balance, tipping it in favor of angiogenic stimulation [62,63]. Conversely, a recent proposal suggests the potential effectiveness of sEVs obtained from bone marrow-derived mesenchymal stem cells in alleviating diabetes-induced retinal injury. This therapeutic effect is attributed to the suppression of Wnt/β-catenin signaling, leading to a subsequent reduction in oxidative stress, inflammation, and angiogenesis [64].

### 2.3. Autophagy and sEVs

Perturbing RPE homeostasis by impairing autophagy promotes inflammasome activation and activates macrophage-mediated angiogenesis, which influences the key features of AMD development [65]. Some authors have demonstrated the link between autophagy and exosome biogenesis in disease. Bhattacharya and collaborators described how GAIP interacting protein C terminus (GIPC) regulates both mechanisms in pancreatic tumor cells [66]. Wang described how autophagy and RPE-derived sEVs work together for the generation of drusen in AMD [67,68]. Increased autophagy enhances tube formation and endothelial cell migration. Moreover, if autophagy is inhibited, endothelial cell migration and tube formation are reduced [69]. Oxidative stress-induced retinal astrocytes modulate proliferation and migration of endothelial cells by increasing retinal astrocytes’ autophagy and the release of sEVs [70].

Increasing evidence indicates that impaired autophagy is associated with angiogenesis, both in the development of the chick embryo [71], where Atg7 plays an important role, as in RF/6A cells, from choroid, where hypoxia-induced autophagy stimulated endothelial growth [72]. Results from our group indicate that inhibiting autophagy in ARPE-19 cells by Atg7 siRNA decreases the expression of VEGFR2 in low-stressed RPE cells. This effect was accompanied by a decrease in the total number of sEVs and the VEGFR2-positive sEVs released from ARPE-19 cells [36]. Interestingly, high doses of the antioxidant curcumin have shown activation of autophagy [73]; these authors did not evaluate sEV release, though their results allow the speculation that, under these conditions, an increase in sEVs could certainly be expected.

Similarly to the methods of other authors, photo-oxidative stimulation with blue-light induced ARPE-19 cells to release sEVs with higher levels of IL-1β, IL-18, and caspase-1 than those unstimulated, whereas the levels of these same factors in these cells were significantly enhanced when treated with sEVs from stimulated cells; furthermore, the NLRP3 mRNA and protein levels were found to be markedly higher in the treated group than in the untreated one [74]. An interesting correlation has been proposed between NLRP3 inflammasome activity and autophagy [75].

The addition of paraquat to ARPE-19 cells somehow mimics the enhanced oxidative burden of the cellular environment in neovascular-AMD. Proteomic analysis of the sEVs released under this condition was performed, and the outcome showed the upregulation of proteins related to the autophagy pathway, such as cathepsin D, similar results to what was observed in the aqueous humor of AMD patients [76].

sEVs released from RPE appear to cross the Bruch’s membrane, reaching and influencing choroidal endothelial cells’ fate. Notably, sEVs originating from retinal astrocytes demonstrate the ability to reduce vessel leakage in a model of AMD, whereas those derived from the RPE fail to prevent leakage from new vessels [61]. Additionally, in another AMD model, it has been proposed that sEVs released by aging RPE cells can enhance autophagy in neighboring cells, potentially contributing to the formation of drusen [67]. Concurrently, another group demonstrated that sEVs derived from mesenchymal stem cells inhibit neovascularization by down-regulating VEGF expression [77]. The interplay between autophagy and sEVs biogenesis is well established, suggesting a connection between these two processes. Notably, under oxidative challenge, ARPE-19 cells exhibit an enhancement in autophagy, an increase in VEGF release, and a higher liberation of sEVs [22,36]. This intricate relationship underscores the interconnected nature of autophagy, VEGF release, and sEVs production in response to oxidative stress in these cells [78,79,80]. RPE might not be the only cellular type of the retina that might release different types of extracellular vesicles as a modulation response, as has been recently described in studies of EV activity within the retina during poly ADP ribosylation in degenerating rd10 [81] and rd1 [82] mice retinae.

### 2.4. Micro RNAs

As mentioned above, sEVs might contain different types of molecules, among which are micro RNA (miRNA) sequences [25], short (around 22 nucleotides), single-stranded RNA molecules that have been proposed as potential therapeutic tools and diagnostic markers in DR [83,84] and also AMD [85,86]. They are found inside the cell as well as the extracellular medium, including plasma. MiRNAs are non coding RNAs involved in transcription and post-transcriptional messenger RNA (mRNA) sequences normally by degrading mRNA transcription or even repressing translational processes. As an example, in RPE samples from elderly humans, it has been described that miR-21 from sEVs that were then taken up by microglia, led to a change in expression of genes in the p53 signaling pathway, implicating microglia in the regulation of aging and AMD [87]. It has been also demonstrated that the injection of up-regulated exosomal miR-17-3p contained in sEVs from human umbilical cord mesenchymal stem cells, reduced blood glucose and HbAlc levels, increased body weight, Hb content and the glutamine synthetase level, decreased the contents of inflammatory factors and VEGF, alleviated oxidative injury, and inhibited retinal cell apoptosis in an experimental mouse diabetes model, through inhibiting signal transducers and activators of transcription 1 (STAT1) [88].

These miRNAs contained in sEVs have in fact been proposed as mediators of homeostasis [89], which, we believe, they might be. In this report, photo-oxidative conditions induced a decrease in the amount, but not in their size, of sEVs production (contrary to what has been reported by others under photo-oxidative stimuli [73]), obtained from whole mouse retina homogenates. Other publications showed, as mentioned above, that other oxidative stimuli, in fact, increase the amount of sEVs production [22,35,36], it might well be that this discrepancy resides in the sEVs collecting procedure. Nevertheless, these authors proposed that retinal health requires optimal levels of sEVs and their cargo; thus, replenishing sEVs loads in the retina may prove to be an efficacious therapy, as mentioned above [39].

Furthermore, some miRNAs have been involved in eye-related vascular proliferation. Initially proposed as a hypothesis, that RPE derived sEVs and their miRNA content could be related to choroidal neovascularization in AMD [90] was relatively quickly confirmed in different experimental models [22,25,36,80,91,92]. Collectively identified as “angiomiRNAs”, they are expressed in endothelial cells, mir-126, mir-210, mir-221/222, among others, and are able to promote or repress angiogenesis (reviewed in [93]). In this sense, bone marrow mesenchymal stem cell-derived sEVs are able to reduce inflammation and angiogenesis, concomitantly with the up regulation of antioxidant enzymes, in diabetes-induced retinal injury [64].

As already mentioned, oxidative stimuli promoted sEVs release from RPE. These sEVs contain different miRNA cargoes with pro-angiogenic properties [25,36]. Recent data from our lab indicate that mir-205 negatively modulates HIF-1a and VEGFA mRNA expression [91]. Under oxidative conditions (high glucose), mir-205 levels significantly decreased, while HIF-1a and VEGFA mRNA expression levels were increased, finally leading to angiogenesis in human umbilical vein endothelial cell (HUVEC) cultures. After adding the antioxidant NAC, mir205 levels were normalized along with HIF-1a and VEGFA mRNA expression levels. Under these conditions, HUVEC cells were unable to produce tubes and branches [91]. In a similar approach, we transfected ARPE-19 cells with a miR-302a-3p mimic and compared the effect of adding sEVs obtained from these cells to HUVEC on tube and vessel formation, with the effect of sEVs from ARPE-19 cells incubated with hydrogen peroxide, and confirmed that miR-302a-3p contained in sEVs can modify VEGFA mRNA expression levels as part of its antiangiogenic features [92]. These results strongly indicate the pivotal role of miRNAs as angiogenic modulators, especially under oxidative conditions.

Figure 1 represents an schematic diagram of the main relationships described in this review that confirm sEVs release and their cargo as mediators in the retinal diseases studied and the different pathways directly or indirectly involved in these oxidative-related pathophysiological mechanisms.

## 3. Conclusions

Unraveling the mechanisms described above may provide new targets for therapy in different diseases, and as they show some common features, knowing, and eventually being able to modify, the cargoes of sEVs could certainly open future directions for the diagnosis and treatment of ocular diseases. It has been described by us and many others that oxidative burden is a triggering mechanism for the release of cell messages in different environments and of course in the retina. Retinal cells release sEVs that may bring signals to other neighboring cells, but also reach blood flow and be investigated in blood samples, provided that adequate markers are used. Their cargo might constitute a signature of what is happening in their cells of origin. Mimicking the configuration of these sEVs, it would ideally be possible to convey the right message in the right place.

## Figures and Tables

**Figure 1 ijms-25-01618-f001:**
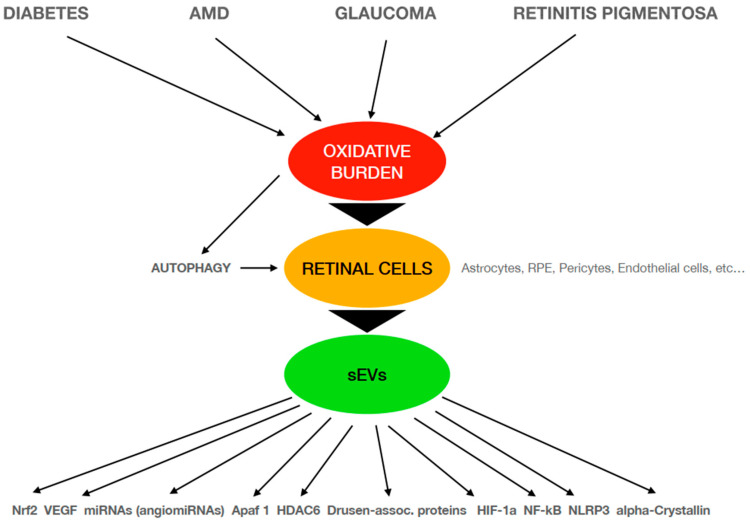
Schematic diagram of the pathophysiological mechanisms involved in different retinal diseases that are related to oxidative conditions. AMD: age-related macular degeneration; sEVs: small extracellular vesicles; RPE: retinal pigment epithelium; VEGF: vascular endothelial growth factor; Apaf-1: apoptosis protease-activating factor-1; HDAC6: histone DeACetylase 6; HIF-1a: hypoxia-inducible factor 1-alpha; NLRP3: NOD-, LRR- and pyrin domain-containing protein 3.

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
