# Peer review of "Small Extracellular Vesicles and Oxidative Pathophysiological Mechanisms in Retinal Degenerative Diseases"

_ijms, 2024, doi:10.3390/ijms25031618_

Round 1

Reviewer 1 Report

Comments and Suggestions for Authors

View attach file.

All the best.

Comments on the Quality of English Language

no comment

Reviewer 2 Report

Comments and Suggestions for Authors

In the article titled “Small Extracellular Vesicles and Oxidative Pathophysiological Mechanisms In Retinal Degenerative Diseases”, Romero and co-workers reviewed the current literature concerning the pathophysiological mechanisms of retinal degenerative diseases analyzing the role of small extracellular vesicles network used by retinal cells to communicate with neighbouring cells. In particular, the authors describe the altered cell-to-cell communication in the retina in the course of pathological conditions with a major emphasis on cell clearing system, mainly the autophagy pathway.

Overall, the present review is well-written and convincing, and the authors have fully achieved the aim set in the introduction. However, some points should be addressed by the authors before considering it for publication:

-       The authors should consider the very recent hypothesis that extracellular accumulation of aggregate of misfolding protein due to autophagy alteration is common in several neurodegenerative diseases. This is the case of Lewy bodies in Parkinson’s disease and dementia with Lewy bodies, plaques and neurofibrillary tangles in Alzheimer's disease. When considering the protein composition of these structures one of the major cargos is alpha-synuclein which is also present within drusen in AMD. Again, the main cell clearing system that is involved in the pigment epithelium is the autophagy pathway, along with the ubiquitin-proteasome system, which is critical to preventing neurodegeneration. The author should also comment on this intriguing aspect of retinal degeneration (please refer to doi: 10.12871/000398292020343). 

-       It would be worth adding a conclusion section that remarks on all the major points addressed in the review paper (like a summary of the take-home message of the paper);

-       There are some grammar mistakes in the paper that the authors should correct. 

Comments on the Quality of English Language

minor editing required

Round 2

Reviewer 1 Report

Comments and Suggestions for Authors

the paper can be published

Reviewer 2 Report

Comments and Suggestions for Authors

The authors have done a great job addressing my concerns.